# Impact of Nutrition, Microbiota Transplant and Weight Loss Surgery on Dopaminergic Alterations in Parkinson’s Disease and Obesity

**DOI:** 10.3390/ijms23147503

**Published:** 2022-07-06

**Authors:** Sevag Hamamah, Andras Hajnal, Mihai Covasa

**Affiliations:** 1Department of Basic Medical Sciences, College of Osteopathic Medicine, Western University of Health Sciences, Pomona, CA 91766, USA; sevag.hamamah@westernu.edu; 2Department of Neural and Behavioral Sciences, The Pennsylvania State University College of Medicine, Hershey, PA 17033, USA; ahajnal@psu.edu; 3Department of Medicine and Biomedical Sciences, College of Medicine and Biological Science, University of Suceava, 7200229 Suceava, Romania

**Keywords:** gut microbiota, Mediterranean diet, Western diet, microbiota transfer, weight loss surgery, Roux-en-Y gastric bypass

## Abstract

Parkinson’s disease (PD), the second most common neurodegenerative disorder worldwide, is characterized by dopaminergic neuron degeneration and α-synuclein aggregation in the substantia nigra pars compacta of the midbrain. Emerging evidence has shown that dietary intake affects the microbial composition in the gut, which in turn contributes to, or protects against, the degeneration of dopaminergic neurons in affected regions of the brain. More specifically, the Mediterranean diet and Western diet, composed of varying amounts of proteins, carbohydrates, and fats, exert contrasting effects on PD pathophysiology via alterations in the gut microbiota and dopamine levels. Interestingly, the negative changes in the gut microbiota of patients with PD parallel changes that are seen in individuals that consume a Western diet, and are opposite to those that adhere to a Mediterranean diet. In this review, we first examine the role of prominent food groups on dopamine bioavailability, how they modulate the composition and function of the gut microbiota and the subsequent effects on PD and obesity pathophysiology. We then highlight evidence on how microbiota transplant and weight loss surgery can be used as therapeutic tools to restore dopaminergic deficits through optimizing gut microbial composition. In the process, we revisit dietary metabolites and their role in therapeutic approaches involving dopaminergic pathways. Overall, understanding the role of nutrition on dopamine bioavailability and gut microbiota in dopamine-related pathologies such as PD will help develop more precise therapeutic targets to rescue dopaminergic deficits in neurologic and metabolic disorders.

## 1. Introduction

Parkinson’s Disease (PD) is a multifactorial, progressive neurodegenerative disease that results from a combination of genetic and environmental factors, as well as various protective and risk factors [1]. At present, PD is the second most common neurodegenerative disorder worldwide, with a prevalence of 0.3% in industrialized countries [2]. The pathological hallmark of PD is a loss of dopaminergic neurons in the substantia nigra pars compacta (SNpc) of the midbrain and associated α-synuclein aggregation in Lewy bodies [3]. Clinical manifestations, such as dyskinesia, postural instability and gait difficulties manifest once 50–70% of dopaminergic neurons are lost within the midbrain [4]. Over the years, it has been well-documented that PD pathophysiology may begin with α-synuclein accumulation within the gut, which ascends to the dorsal motor nucleus of the vagus via vagal afferents, resulting in the impending degeneration of dopaminergic neurons [5,6,7]. Due to the gastrointestinal involvement with the onset of PD, numerous studies have examined the role of nutrition and associated changes in gut microbiota composition on dopamine bioavailability [8,9]. For example, the Mediterranean diet has been significantly associated with protection from early onset of PD [10], while recent studies show novel findings linking high intake of yogurt and fermented milk, animal-based fats and sweet snacks to increased risk of PD development [11]. The protective effect of the Mediterranean diet is attributed, in part, to increased diversity and favorable enterotypes of gut microbiota, such as a decrease in the Firmicutes/Bacteroidetes ratio, which has been associated with many disease states when elevated [12,13]. Additionally, certain gut microbial species are associated with neurotransmitter availability and can exert neuroprotective effects against PD or exacerbate its pathophysiology [14]. These complex processes are regulated by crosstalk between the gut and the brain via the so-called microbiota–gut–brain axis. The microbiota–gut–brain axis communicates bidirectionally through vagal afferents, short-chain fatty acids produced by gut microbes, the immune system and the hypothalamic–pituitary–axis, all of which can be modulated by dopaminergic activity and dietary interventions [15]. As such, therapeutic methods that restore “ideal” gut microbial composition have been utilized to optimize reciprocal gut–brain communication and rescue dopaminergic deficits in the central nervous system and in the periphery. To date, the literature focuses primarily on the use of probiotics in the treatment of neurodegenerative disorders and their positive effects on dopamine bioavailability. Promising evidence includes probiotic mediated increases in dopamine [16], as well as its metabolites, including 3,4-Dihydroxyphenylacetic acid (DOPAC) and Homovanillic acid (HVA) [17]. Further, probiotics are shown to increase other aspects of neurodegenerative disease, such as improvements in sleep quality, alleviation of anxiety, gastrointestinal symptoms, reduction of oxidative stress and anti-inflammatory effects [18,19]. However, other therapeutic modalities have also been implicated in this regard, although not currently as utilized in practice when compared to probiotics in the context of restoring dopaminergic deficits in neurodegenerative disorders. For example, fecal microbiota transplant (FMT) and weight loss surgery (WLS) have been linked with increased dopamine concentrations [20], dopamine receptor availability [21,22], inhibition of reward-related dopamine signaling [23] and suppression of dopaminergic neuron loss [24], to name a few.

In this review, we explore these relationships by first presenting emerging evidence demonstrating the relationship between nutrition, gut microbiota, and dopamine bioavailability. Then, we briefly describe the changes seen in gut microbial composition profile following dietary interventions and dopaminergic neuronal loss as seen in PD. Lastly, we present evidence linking PD with obesity and therapeutical approaches, such as FMT and WLS, to rescuing the dopaminergic deficits that are characteristic of PD and obesity.

## 2. Changes in Nutrients, Dopamine, and Microbiota

Diet provides natural sources of substrates and precursors that contribute to neurotransmitter biosynthesis and the maintenance of normal brain function. Consumption of optimal levels of carbohydrates, fats and proteins is linked to the bioavailability of biogenic amines such as dopamine, serotonin and norepinephrine, which drive critical physiological processes in the body. The dopaminergic system, in particular, plays a significant role in regulating satiety, food-seeking behaviors, and the motivational drive to eat [25,26]. It is also known that gut microbiota plays significant roles in modulating processes that influence dopamine bioavailability via gut-brain signaling [14]. Different dietary patterns, such as increased intake of animal meat or protein and plant-based carbohydrates, influence the composition of gut microbiota. More specifically, protein-based and animal fat diets have been associated with an abundance of *Bacteroides* in the gut, while carbohydrate-rich plant diets are linked with species within the *Prevotella* genus [27]. Recent findings have shown that these two bacterial enterotypes can have varying effects on dopaminergic signaling [28]. For example, introduction of *Bacteroides uniformis* in mice through FMT resulted in increased striatal dopamine transporter binding (DAT), while *Prevotella copri* was negatively correlated with DAT binding. DAT is a critical modulator of dopaminergic tone within the CNS by allowing dopamine recycling and storage within vesicles for proper synaptic transmission [29]. Therefore, altered striatal DAT expression has been linked with neurologic diseases that are characterized by dopaminergic neuronal degeneration as seen in PD [30,31]. Furthermore, carbohydrate-based diets have been associated with relative abundances of *Bifidobacterium* and *Ruminococcus* [32], both of which have strong links to dopamine neurotransmission and disease processes (Figure 1) [33]. Similarly, some low-fat calorie sweeteners have also been associated with increased concentrations in *Bifidobacterium* genus. *Bifidobacterium* is correlated with a myriad of beneficial effects in the human host through alterations to neurotransmitter profiles, including dopamine [34], and is a common bacterial genus used in many probiotics. Additionally, foods such as kiwifruit have been shown to increase *Bifidobacterium* species within in-vitro fecal fermentation models, which also enhance the production of key dopamine precursor l-dihydroxyphenylalanine (L-DOPA). Taken together, these findings provide strong evidence for the bio-aminergic potential of certain food groups while indicating the importance of gut microbiota and its effects on dopaminergic signaling. As a result, dietary interventions have been frequently used in treating neurological diseases. In the following subsections, we will further discuss the effects of the main food groups and their impact on gut microbiota and dopaminergic pathways.

### 2.1. Carbohydrates and Their Effects on Microbiota and Dopamine

Gut microbes have the capability to metabolize complex carbohydrates present in indigestible and dietary fibers starches to produce neuroactive short-chain fatty acids (SCFA). SCFA-producing bacteria, such as *Lactobacillus rhamnosus,* are shown to increase vagal afferent firing frequency, inducing activation of dopaminergic neurons, and dopamine concentrations in the brain [41,42]. Therefore, specific dietary carbohydrates serve important but varying roles in gut-brain signaling and dopamine bioavailability.

For example, fructo-oligosaccharides (FOS) found in bananas, soybeans, asparagus, and other foods are known to serve as a growth factor for beneficial bacteria such as *Bifidobacterium* and *Lactobacillus* [35]. FOS are not digested until reaching bacteria in the colon, where they have been shown to stimulate the growth of symbionts, decrease appetite via GLP-1 [43] and improve behavior deficits [44]. When given in conjunction with *Bifidobacterium*- and *Lactobacillus*-containing probiotics, FOS stimulated increased SCFA concentrations to alleviate dopamine metabolism disorders in patients with autism spectrum disorder (ASD) [45]. In support of the positive findings of FOS and dopamine concentrations, supplementation of FOS and xylo-oligosaccharides (XOS) increase dopamine levels in the cortex of rodents [46]. Additionally, prebiotic FOS supplements have been used to reverse mesolimbic-induced hedonic control of feeding after 2 months of exposure to high-fat-high-sugar diets [47]. The findings suggest that FOS supplements increase genes involved with dopamine signaling in the nucleus accumbens (NAc) and hypothalamic orexigenic markers while decreasing palatable food consumption. These studies support the positive effects of FOS dietary interventions on dopamine concentrations as well as dopamine-related physiological processes and behaviors.

Other dietary carbohydrates such as sugars have roles in modulating dopaminergic activity and gut microbial enterotypes. For example, sugars increase dopamine levels in a dose-dependent manner in reward-associated centers of the brain such as the NAc (Figure 2) [48,49]. However, these dose-dependent increases in brain dopamine levels are only associated with acute consumption. Chronically administered high-sugar diets in rats showed reduced levels of D2R expression and decreased striatal dopamine levels [50]. It has also been shown that withdrawal of excessively high sucrose diets further reduced dopamine levels in the NAc and induced anxiety-like behaviors [51]. Additionally, sucrose was found to decrease the rate-limiting enzyme of dopamine synthesis, tyrosine hydroxylase (TH), in the medial prefrontal cortex [52]. Overall, these findings support a strong correlation between long-term sugar diets and their negative impact on dopamine levels in key regions of the brain.

Furthermore, the metabolic changes induced by high sugar diets result in altered dopaminergic signaling and gut microbial profiles. For example, a high fructose corn syrup (HFCS) diet induced glucose dysregulation that occurred in parallel with reduced dopamine release [53]. The high-sucrose diets that induced obesity and hyperglycemia significantly enhanced D2 receptor protein level while reducing D2 receptor mRNA and binding affinity in the striatum [54]. Similarly, the negative effects of high-sugar diets and resulting metabolic dysregulation have been observed through unfavorable gut enterotypes. Most notably, the introduction of high-glucose or high-fructose diets lowers gut microbial diversity, through decreases in Bacteroidetes and increases in Proteobacteria [36]. The resulting increase in the Firmicutes/Bacteroidetes ratio contributes to increased gut permeability resulting in lipopolysaccharide-induced endotoxemia. These alterations in gut microbiota lead to increase cytokine production which can influence neural activity by crossing permeable regions of the brain to influence microglial activation. Excessive microglial activation via gut microbial contents induces neuroinflammation via production of pro-inflammatory cytokines such as IL-1 and Tumor Necrosis Factor-α (TNF-α) leading to irreversible CNS damage and neurodegenerative pathology [55]. Furthermore, 1-methyl-4-phenyl-1,2,3,6-tetrahydropyridine (MPTP), a neurotoxin that degrades dopaminergic neuron to induce PD in mouse models, has been shown to be dependent on pro-inflammatory cytokine receptors such as the TNF-α receptor [56]. Therefore, inflammation linked to gut microbiota is a major component of the development of the dopaminergic neurodegeneration seen in PD. These findings suggest potential therapeutic role of procedures such as weight loss therapy in ameliorating effects of metabolic disorders and restore dopaminergic deficits through alteration of gut microbial profiles. These effects will be discussed in a later section.

### 2.2. Omega-3 Poly-Unsaturated Fatty Acids, Microbiota and Dopamine

Omega-3 poly-unsaturated fatty acids (PUFA), found in fish, plant oils and nuts, have also been studied in the context of dopamine and neurological disorders. Dietary omega-3 deficiency is correlated with a unique sensitivity in adolescents, leading to enhanced dopamine availability in the striatum [63]. However, in adults, the detrimental effects of omega-3 deficiency become apparent over the course of an individual’s lifetime, indicating age-dependent changes in dopamine neurotransmission. This is supported by findings showing prolonged omega-3-depletion, until adulthood, in rodent models of impaired striatal antioxidant resources and brain-derived neurotrophic factor (BDNF) distribution [57]. BDNF has been shown to be neuroprotective against dopamine degeneration [64]. Thus, depletion of BDNF would further worsen the degeneration of dopaminergic neurons and increase the risk of PD development. Supplementation with fish oil in 6-OHDA neurotoxicity-induced PD models of mice reduced loss of neurons and nerve terminals in the SNpc and striatum, respectively, by reducing microglia and astrocyte activity [58,65]. As mentioned, neurotoxicity-induced excess in microglial activation results in neuroinflammation contributing to dopaminergic damage [55] and PD pathophysiology. As such, the reduction in microglial activity supports the neuroprotective effects of omega-3 PUFA against dopaminergic neuron degeneration.

Recent studies demonstrate that fish oil from omega-3 PUFA prevents against oxidative damage in the prefrontal cortex via alterations in dopamine neurotransmission [59]. In this study, fish oils exhibited neuroprotective effects, by reducing dopamine 1 receptor (D1R) and dopamine 2 receptor (D2R) levels in response to elevation seen in amphetamine-induced rats (Figure 2). This response decreases excess dopamine signaling through the mesolimbic and nigrostriatal pathways, therefore reducing reward-seeking behaviors and neurotoxicity [66]. Although Metz et al. [59] observed no changes in TH activity in their study, a more recent study reports beneficial effects of omega-3 PUFA on TH+ neurons. Doubling dietary intake of omega-3 PUFA increased TH+ cells in the ventral tegmental area (VTA) and NAc, resulting in more mobility behaviors in mice [67].

Omega-3 PUFA supplementation and its resulting changes in gut microbiota have a significant role in the beneficial effects seen on dopamine neurotransmission through gut–brain signaling [68]. Normalization of the Firmicutes/Bacteroidetes ratio is one mechanism by which omega-3 PUFA prevent neurologic deficits [69,70]. In a recent study, health-promoting bacteria belonging to Bacteroidetes, along with *Lactobacillus,* was found to be significantly elevated after omega-3 PUFA supplementation (Figure 1) [37]. As mentioned, *Lactobacillus* is commonly used in probiotics and is associated with positive effects on dopamine concentration. For example, *Lactobacillus*-containing probiotics resolved social deficits from impaired dopaminergic reward systems. Interestingly, the probiotic initiated increases in oxytocin in the paraventricular nucleus of the hypothalamus, which, in turn, increased the excitability of dopaminergic neurons in the VTA and synaptic plasticity [71]. This effect required an intact vagus nerve, as *Lactobacillus reuteri* supplementation indirectly attenuated impaired dopaminergic reward systems by stimulating oxytocin release in vagotomized mice. Furthermore, *Lactobacillus reuteri* and *Bifidobacterium adolescentis* probiotic therapy reduced lipopolysaccharide (LPS) production and colonic inflammation while increasing hippocampal BDNF levels. The probiotic also prevented activated microglia from permeating into the hippocampus [72]. Encouraging findings were also seen in another study that introduced *Lactobacillus plantarum* to stress-induced/sleep-deprived mice [73]. Supplementation of *Lactobacillus plantarum* probiotic decreased brain monoamine oxidase type B (MAO-B) levels. MAO-B is an enzyme that, converts dopamine into metabolites that can be excreted by the body. Therefore, *Lactobacillus plantarum*-mediated decreases in MAO-B serve as a marker of decreased dopamine breakdown and increased bioavailability. Additionally, intestinal BDNF expression was elevated in the hippocampus of the stress-induced mice. Taken together, these findings support the beneficial effects of omega-3 PUFA on dopamine bioavailability through optimization of gut microbiota.

### 2.3. Dietary Protein, Microbiota and Dopamine

Consumption of proteins have been associated with varying effects on dopamine and gut microbiota. Proteolytic fermentation of proteins by gut microbiota yields beneficial free amino acids and SCFAs, while also producing ammonia and p-cresol, which have been shown to have negative effects such as DNA damage and cell-cycle alterations [74]. However, the effects of derived metabolites are dependent on the protein type that leads to a myriad of effects on gut bacterial composition and cognitive functions. For example, protein supplements have been associated with increased Bacteroidetes, but decreased health was related to *Roseburia, Blautia* and *Bifidobacterium* species (Figure 1), indicating potential negative impact of long-term protein intake [75]. Similarly, animal-fat proteins have been shown to decrease *Bifidobacterium* and SCFA production [38] in the gut, as well as increase *Desulfovibrio* species [39], contributing to inflammatory bowel states. In contrast, plant-based proteins are shown to stimulate SCFA production, through promoting increased abundances of commensal bacterial species such as *Lactobacillus* and *Bifidobacterium* [40].

Protein deprivation has been shown to decrease cognitive function and cause behavioral abnormalities in mice models [76]. For example, 2 months of low protein diet in mice resulted in agitation and hyperactivity while decreasing dopamine concentrations and brain neurotransmitters [77]. These behavioral changes were partially restored by administration of essential amino acids from high-protein diets. The rate of dopamine synthesis is dependent on amino acids, particularly tyrosine, which is the precursor of the rate-limiting step in the biosynthesis pathway. As such, both daily and long-term protein content influences tyrosine levels, and dopamine concentrations [60]. Additionally, acute protein intake was found to reduce post-meal cravings and elicit sustained increases in HVA concentration in obese adolescents [61]. HVA is a major dopamine metabolite and is an indicator of central dopaminergic activity. These findings demonstrate that proteolytic fermentation of protein diets yields amino acids that contribute to dopamine synthesis and bioavailability in reward centers of the brain. Furthermore, a recent study showed that protein restriction in adolescents was associated with decreased dopamine in NAc, a major interface of the mesolimbic reward pathway of the brain. These findings coincide with those of Hoertel et al., who concluded that adolescent individuals exhibit higher HVA and brain dopamine levels (Figure 1) [62]. However, it seems that changes in brain dopamine availability is age dependent, since adult mice were shown to have higher dopamine levels compared to adolescent mice in the NAc, following protein deprivation [62].

Although some studies have associated a low protein diet (LPD) with decreased dopamine bioavailability [77], low protein diets have been found to have beneficial effects in the treatment of PD. For example, LPD ameliorates motor symptoms in mice by reducing the neurodegeneration of TH+ neurons [78]. The beneficial effects on TH+ neurons is due to concurrent reductions in activated microglia, proinflammatory cytokines and abnormal protein inclusions [79]. In addition, high protein diets lessen the therapeutic effects of Levodopa, the most effective medication for PD, by impairing its absorption. Taken together, these findings support adherence to a LPD in conjunction with Levodopa to improve the symptomology and neurodegenerative progression of PD.

## 3. Dietary Intervention, Gut Microbiota, Dopamine and PD

The Mediterranean Diet (MD) has been extensively studied in recent years due to its beneficial association with neurodegenerative diseases such as PD and Alzheimer’s disease [80]. The MD is comprised of minimally processed plant-based foods, such as whole-grain cereals, wheats, nuts, seeds, PUFA, fruits and vegetables which, collectively, are shown to be rich in antioxidants and anti-inflammatory agents [81]. Consumptions of high fiber diets, such as wheat bran, result in upregulated BDNF production in peripheral tissues [82]. Other beneficial effects of the high fiber-containing MD include alleviating inflammation, increasing insulin sensitivity, and improving intestinal barrier integrity [83,84]. Furthermore, these favorable dietary nutrients confer protective effects, both in delaying the onset of PD and reducing the symptomology of those currently affected. A recent cross-sectional study supports the protective effects of MD on delaying the onset of PD up to 17.4 years in females and 8.4 years in males [85]. These sex-dependent differences were attributed to females being more adherent to the diet throughout the course of the study; however, other sex-dependent and lifestyle factors exist. Another population-based study that followed dietary habits since 1991 found a 29% reduced risk for developing PD for middle-aged individuals that were adherent to the MD. Furthermore, starting the MD in patients currently diagnosed with PD augmented overall cognitive function with marked improvements in executive function, attention, concentration, and active memory [10]. In addition, short-term MD adherence decreased constipation symptoms and altered fecal microbiota, restoring levels of *Roseburia* and decreasing *Bilophila* over a 5-week period (Figure 3) [86]. The implications of the MD in gut microbial composition are significant with studies showing up to 57% structural variation in gut microbiota as a direct result of the diet [87]. Gut microbiota sequencing has associated MD adherence with a decreased ratio of Firmicutes/Bacteroidetes [88]. More specifically, the MD increases the relative abundances of species, including, but not limited to, *Faecalibacterium prausnitzii, Eubacterium* and *Roseburia.* Collectively, the positive impact of these microbial taxa has been attributed to significantly upregulated in production of SCFA. The MD also selects against the growth of unfavorable species, such as *Ruminococcus torques* and *Coprococcus comes,* which are associated with the synthesis of harmful metabolites including acetone, p-cresol and ethanol.

On the other hand, the Western Diet (WD) consisting mainly of fats, sugars and animal proteins has essentially opposite effects to MD. For example, a recent case-control study showed that consuming large lifetime quantities of red meat, a major component of the WD, is positively associated with PD [89]. As mentioned, consumption of dietary proteins derived from animal sources leads to decreased *Bifidobacterium* and an overall gut enterotype that promotes inflammation [39]. Furthermore, introduction of a high-fat/high-sugar diet mimicking the WD in mice increased *Escherichia coli* and *Ruminococcus torques* while decreasing *Roseburia* and *Eubacterium* (Figure 3) [90,91]. These negative impacts can be largely attributed to the availability of these nutrients to specific gut microbial species in the WD. With a lesser consumption of dietary carbohydrates, animal proteins are proteolytically degraded by gut microbiota leading to harmful metabolites and growth of unfavorable gram-negative microbes [92]. Moreover, transient increases in *Ruminococcus* species, that can result from ingestion of dietary substrates found in the WD, are associated with inflammatory episodes present in inflammatory bowel disease [93] and production of pro-inflammatory cytokines such as TNF-α [94]. These findings were supported by another study showing that WD feeding of mice induced systemic inflammation through alterations of LPS responses and NLRP3 inflammasome innate immune reprogramming [95], both of which are significantly associated with PD development. Activation of LPS, a major endotoxic component of the outer membrane in gram-negative bacteria, by Toll-like receptor-4 (TLR-4) leads to the assembly of the NLRP3 inflammasome and production of abundant amounts of interleukins. Chronic LPS-mediated endotoxemia has been shown to reduce striatal dopamine and D2 receptor binding [96]. Additionally, excess LPS contributes to the production of reactive oxygen species (ROS), resulting in dopaminergic neuron loss, reduced TH activity in the SN and increased microglial activation [97]. The NLRP3 inflammasome has been found to have similar detrimental effects on PD development through excessive microglial activation resulting in sustained neuroinflammation. This was shown through a significant upregulation of the NLRP3 inflammasome, localized mostly to microglia in the substantia nigra in rodent PD models [98]. These findings also suggest that inflammasome inhibition mitigates motor deficits, nigrostriatal dopaminergic degeneration and α-synuclein accumulation. Taken together, there is strong evidence supporting WD-induced inflammation and its role in PD pathogenesis.

In addition to their effects on PD, several studies have evaluated the effects of MD and WD diets on dopamine and other related pathologies. For example, the WD and MD demonstrated opposite effects on morphine addiction. WD exaggerated anxiety and stress-behaviors via exacerbation of the hypothalamus-pituitary-adrenal (HPA) axis leading to morphine withdrawal symptoms, while MD prevented morphine re-instatement [99]. Dysregulation of the HPA axis is shown to have negative consequences on gut microbiota primarily through the upregulation of pro-inflammatory cytokines such as TNF-α. IL-6 and IL-1β [100]. Likewise, the supplementation of beneficial gut microbes has been found to normalize the stress response and maintain homeostasis of the HPA axis [101]. Therefore, introduction of favorable gut microbes via MD diet adherence can restore HPA axis integrity and promote favorable effects on morphine re-instatement. Furthermore, a WD modified dopamine neurotransmission by increasing DAT and D2R immunoreactivity and decreasing D1R in the NAc, in contrast to the MD, which preserved dopamine mesolimbic neuroplasticity. These findings suggest that the MD generally confers positive effects on dopamine bioavailability, while the WD can be harmful.

## 4. Changes in Microbiota in PD

Extensive metagenomic sequencing of the parkinsonian gut has shown trends in the relative enrichment and decreased abundance of specific microbial species [102]. Overall, some of the notable alterations in gut microbial composition in PD include increases in *Ruminococcus, Akkermansia*, *Eggerthella,*
*Enterobacteriaceae**, Catabacter, Shigella, Lachnospiraceae, Streptococcus, Eschiria, Enterococcus* with reductions in the concentrations of *Bacteroides, Coprococcus, Prevotella, Faecalibacterium* and *Clostridium* species. The development of pathognomonic symptoms of PD has been directly associated with some of these gut microbial alterations. For example, an abundance of *Enterobacteriaceae* is linked with motor symptoms of PD, specifically abnormal gait and postural instability [103]. Notable trends in the gut of persons with PD generally favor the growth of unfavorable bacteria that are involved in PD pathophysiology, such as *Enterobacteriaceae,* while reducing microbes that are neuroprotective against dopaminergic degeneration. Furthermore, gastrointestinal dysfunctions such as intestinal dysmotility correlate with gut microbial changes, which can be significantly attributed to elevated Firmicutes/Bacteroidetes ratio [104]. For example, *Prevotella,* a species belonging to the phylum Bacteroidetes, has been shown to be decreased in the parkinsonian gut, therefore elevating the Firmicutes/Bacteroidetes ratio and contributing to intestinal dysmotility [105]. In the gut, *Prevotella* exerts neuroprotective properties against TH-containing dopaminergic neuronal degeneration. This protective effect is mediated by the secretion of hydrogen sulfide into the gut lumen as seen in rodents [106]. These findings are supported by studies showing that *Prevotella*-enriched enterotypes are associated with significantly lowered constipation severity when compared to Firmicute-enriched enterotypes [107]. Further, sequencing of gut microbiota following WD adherence revealed limited prevalence of *Prevotella* [108] which correlates closely with the development of PD and its associated symptoms.

*Prevotella* and *Enterobacteriaceae* are two microbial taxa that have been highly associated with the development of characteristic symptoms of PD; however, other microbial changes exist. Overall, taxonomic alterations include elevations in mucin-degrading species such as *Akkermansia* and *Ruminococcus* [109], two genera that increase gut permeability at high concentrations. Moderate to low levels of *Akkermansia* were shown to have beneficial effects in the maintenance of intestinal barrier integrity; however, abnormal elevations increase leakiness of the gut barrier through downregulation of tight junction proteins [110]. As such, intestinal leakiness permits endotoxins such as LPS to escape into systemic circulation to cause neuroinflammation. This has also been shown through microscopic examination of LPS and α-synuclein demonstrating LPS induces amyloidogenesis through rapid nucleation events and α-helical intermediates, further worsening PD pathophysiology [111]. In this context, diet also becomes important, as foods that increase these mucin-degrading species, such as WD-induced elevations in *Ruminococcus,* contribute to the unfavorable gut microbial profile as well. Furthermore, the parkinsonian gut exhibits decreased abundances of *Roseburia* and *Eubacterium* spp., corresponding to changes seen by individuals that primarily consume a WD. As mentioned in Section 3, the mechanisms by which WD-induced elevations in *Ruminococcus* as well as decreases in *Roseburia* and *Eubacterium* involves production of ROS, pro-inflammatory cytokines, excessive microglial activity and neuroinflammation [94,97]. Taken together, these physiologic changes contribute to dopaminergic neuron loss and PD pathogenesis. Conversely, MD adherence can restore *Roseburia* potentially playing a role in delaying the negative manifestations in the parkinsonian gut.

## 5. The Link between Parkinson’s Disease and Obesity

Obesity, characterized by excessive body fat accumulation and adipose tissue dysfunction, has been shown to have significant associations with neurodegenerative disease [112,113]. PD and obesity share commonalities in their pathophysiologic mechanisms, including a loss in dopaminergic neurons accompanied by reductions in brain dopamine levels [114,115]. These shared characteristics were first described in a study showing decreased availability of striatal D2R in obese patients [116]. It was suggested that increasing caloric intake may serve as a compensatory mechanism to decreased reward pathway activation in these patients. Therefore, reduced striatal D2R is a potential mechanism by which dopamine can worsen obesity. It was also found that chemical-induced neurotoxicity in dopaminergic neurons was enhanced in obese mice, indicating that obesity can be a risk factor for accelerating disease-induced neurodegeneration as well [117]. Further, a cross-sectional study, examining the BMI of PD patients versus controls, reported an increased prevalence of obesity in the PD group as measured by statistically significant elevations in BMI [118]. Due to these findings, studies have more thoroughly explored the mechanisms by which long-standing obesity can induce or worsen PD pathophysiology, primarily through causing neuroinflammation and secondarily by upregulating key gut hormones such as GLP-1.

Adipose tissue dysregulation in obesity triggers release of free fatty acids which, in turn, leads to a systemic pro-inflammatory state through activation of inflammatory mediators such as TNF-α, IL-6, IL-1 and IL-12 [119,120]. These inflammatory mediators, particularly IL-6 and IL-12, enhance cognitive decline and reduce processing speed [121,122]. More specifically, a combination of these free fatty acids and pro-inflammatory cytokines can enter the circulation, reaching astrocytes and microglia within the brain to induce neuroinflammation. For example, high-fat diet (HFD)-induced obesity in mice resulted in elevated markers of astrocytes (GFAP) and microglia (Iba1) in the pre-frontal cortex [123]. Furthermore, a HFD has been shown to increase cortical microglial release of TNF-α [120]. It has been documented that neuroinflammation in PD works through a similar mechanism, with α-synuclein aggregation states increasing microglia-induced TNF-α secretion as shown in transgenic rodents [124]. Resulting neurotoxicity, in turn, can promote a positive feedback mechanism where TNF-α further reactivates microglia leading to further TNF-α release, contributing to neuroinflammation and dopaminergic neuron loss [125]. These neuroinflammatory pathways have been targeted by therapeutic modalities, such as fecal microbiota transplant, which have been shown to suppress this neuroinflammation by reducing glial cell activity and TNF-α levels in the brain [24]. In addition to excessive microglial activation, astrogliosis induced by obesity has also been indicated in PD development. Astrocytes are involved in PD pathophysiology, playing a role in the early tissue response, α-synuclein accumulation and neurodegeneration [126]. In a recent study, an obese mouse model generated through 5 months of HFD feeding exhibited increased astrogliosis in the SNpc and striatum [127]. TH staining from these same mice revealed reduced nigrostriatal dopaminergic neurons. A more recent study supports these findings, showing that obesity triggered reductions in TH levels in the VTA and is associated with insulin resistance, increased TNF-α levels, oxidative stress, astrogliosis and microgliosis [128]. Interestingly, no differences in α-synuclein levels were found, suggesting that reductions in TH activity were due to α-synuclein-independent mechanisms such as neuroinflammation. Taken together, these findings suggest that obesity-induced inflammatory markers can stimulate brain structures such as microglia, leading to dopaminergic neuronal degeneration and development of PD.

Another proposed mechanism linking obesity to PD pathophysiology is through its role in promoting insulin resistance. Emerging evidence in recent years has linked and type 2 diabetes mellitus (T2DM) and its insulin dysregulation to worsen PD outcomes [129]. GLP-1 is a gut hormone that serves important roles in the development of T2DM and PD. GLP-1 analogs and GLP-1R agonists have been shown to enhance synaptic membrane expression of DAT, and have neuroprotective effects against dopaminergic neuron degeneration [130,131]. In T2DM, GLP-1 levels are decreased, permitting glucagon action, worsening blood glucose, and increasing insulin resistance [132]. GLP-1 receptor (GLP-1R) agonists, such as liraglutide and exenatide, have demonstrated beneficial effects in both T2DM and PD, including enhancing dopamine midbrain function and promoting glucose-dependent secretion of insulin [133]. Further, GLP-1 exerts anti-inflammatory and anti-oxidant effects to promote nerve cell differentiation and inhibit neuroinflammation, both of which are key factors in PD pathogenesis [134]. As such, adequate GLP-1 concentrations is central to preventing neuroinflammation mediated dopaminergic neuron degeneration in the midbrain. It has also been shown that gut microbiota serve an important role in this mechanism via the production of SCFA such as butyrate. Butyrate increases GLP-1R expression in the brain, while also stimulating enteroendocrine cell production of GLP-1 [131]. Therefore, butyrogenic species enhance the body’s GLP-1 levels to exert positive effects on dopamine concentrations. As such, dietary interventions that favor the production of beneficial gut microbiota can naturally restore GLP-1 concentrations. For example, the increase in butyrogenic gut microbiota seen in individuals adherent to the MD stimulates GLP-1 secretion via free fatty acid receptor 2, an SCFA receptor present in the gut (FFAR2) [135]. Restoring adequate GLP-1 levels can lessen insulin resistance, increase BDNF levels in the brain through vagal stimulation, and reduce neuroinflammation and oxidative stress [134,136], further supporting the beneficial role of MD in neurodegenerative pathology. Along with dietary interventions, therapeutic modalities that treat obesity and modify gut microbiota, such as weight loss surgery and microbiota transplants, are becoming increasingly studied in the context of PD. In the following sections, we will discuss how these modalities can restore dopamine deficits to delay or protect against PD pathophysiology.

## 6. Strategies to Restore Dopamine and PD Deficits

Due to the intricate role of gut microbes in human health, alterations of the normal gut microflora can have serious implications in pathological processes. As a result, therapeutic methods that restore optimal gut microbial composition have been utilized. In recent years, FMT and WLS have become increasingly popular in treating prevalent illnesses such as metabolic syndrome, obesity and T2DM, as well as gastrointestinal and neurodegenerative diseases. Both FMT and WLS exert a myriad of beneficial effects by exerting physiological and metabolic changes within the gastrointestinal tract, including modification to the gut microbiota composition and functions. Interestingly, changes in gut microbial composition resulting from these therapeutic methods can rescue neurotransmitter deficits both in the central nervous system and in the periphery. For example, transplantation of fecal microbiota contents of schizophrenic patients into GF mice resulted in upregulation of basal extracellular dopamine levels in the prefrontal cortex [20]. Similarly, fecal microbiota transplant from donors who received post-Roux-en-Y gastric bypass (RYGB) surgery resulted in increased DAT availability in the striatum of recipients [28]. Interestingly, oral butyrate treatment reduced DAT binding, suggesting that post-RYGB benefits on restoring dopamine functions may include microbial mechanisms other than SCFAs. Further, the effects of HFD on dopamine in obese mice were restored via RYGB, through normalization of dopamine levels, D1R, D2R and DAT [137]. The following sections expand upon the potential therapeutic uses of FMT and WLS in rescuing dopaminergic and PD deficits.

### 6.1. Fecal Microbiota Transplant

Fecal microbiota transplant (FMT) is a therapeutic method currently approved to treat recurrent *Clostridioides difficile* infection. It has also been used in clinical and preclinical studies examining microbiota effects in several pathologies including cardiovascular, metabolic and neurological disease [138,139,140,141,142]. In essence, fecal microbiota is transplanted from one organism to another with the aim of restoring normal gut microbial composition. As a result, the newly introduced gut microbiota exert their effects to rescue neurological deficits through gut-brain communication [28,143]. Metabolites from dietary substrates and restoration of dopamine deficits have also been implicated in this process. For example, abnormal social behaviors induced by p-cresol, a harmful byproduct of proteolytic degradation of animal proteins, were restored through FMT [144]. In this study, p-cresol treated mice exhibited selective taxonomic changes to gut microbial composition, notably increasing genus and species belonging to the *Clostridiales* order. These gut changes were associated with impaired central dopamine activity specifically in the VTA and gave rise to social behavior deficits. FMT of normal microbiota into the p-cresol treated mice normalized p-cresol mediated effects, increased VTA dopamine neuron excitability and restored social deficits. Furthermore, FMT ameliorated autism-like behaviors in EphB6-deficient mice, which is a candidate gene deficiency seen in ASD [145]. The positive gut microbiota-mediated effects after microbiota transplantation demonstrated increased serum vitamin B6 and restoration of dopamine deficits. Interestingly, vitamin B6 supplementation did not restore social deficits, supporting the notion that direct modification of the gut microbiota via FMT was the mechanism by which vitamin B6 positively impacted dopamine levels. It has also been shown that transplantation of microbiota from patients with schizophrenia can induce similar behavioral abnormalities in mice [20]. At the same time, FMT increased extracellular dopamine levels in the prefrontal cortex, which is tied to some of the adverse symptoms seen in schizophrenia, such as hallucinations and delusions. In addition, another study demonstrated significant disruptions to monoamine neurochemistry, showing increased levels of HVA/dopamine ratio which indicates increased dopamine turnover (See Figure 3) [146]. Overall, these studies provide strong evidence for the beneficial role of FMT on restoring gut microbial composition and dopamine deficits.

Several studies have supported FMT as a potential therapeutic method for PD. In a recent study examining the neuroprotective effects of microbes on MPTP-neurotoxicity-induced mice, FMT restored striatal dopamine concentrations, suppressed loss of dopaminergic neurons and decreased neuroinflammation [24]. In addition, FMT was shown to decrease expression of TLR4 and pro-inflammatory cytokine, TNF-α, and lowered LPS binding. Thus, subsequent pro-inflammatory effects of LPS were attenuated leading to reduced neuroinflammation that is characteristic in PD. In a similar study, FMT from mice undergoing a fasting-mimicking diet (FMD), into MPTP-induced mice, mitigated striatal dopamine loss [147]. The FMD, which consisted of 3 days of fasting and 4 days of refeeding, has been shown to change markers of aging and cell protection, enhance cognitive function and reduce oxidative damage and inflammation [148]. The lessened dopaminergic neuron loss observed in the study was accompanied by reduced microglial and astrocyte activity and increased BDNF expression [147]. As mentioned, excessive microglial activity is a major initiator of neuroinflammation through production of TNF-α and IL-1β. Through suppression of these pro-inflammatory cytokines, damaging effects of neuroinflammation on dopaminergic neurons are lessened.

Moreover, DAT expression is altered in patients with PD due to the degradation of dopaminergic neurons, an effect that was rescued by the introduction of *Bacteroides uniformis* via FMT in mice [28]. Therefore, it is plausible that, to a certain degree, some microbes such as *Bacteroides uniformis* through FMT can mitigate DAT deficits present in PD patients. Additionally, FMT improves intestinal dysmotility commonly seen in PD. When FMT was performed in patients with slow-transit constipation, stool consistency and colonic transit were enhanced after treatment, an effect that display superior efficacy than other gut-microbial altering methods, such as oral laxatives and probiotics [149]. Taken together, these findings are encouraging, and demonstrate the versatility of FMT as a therapeutic method and support its beneficial effects on restoring deficits associated with dopaminergic pathophysiology. Nevertheless, more research is needed before implementing this method as a treatment for neurodegenerative diseases in humans.

### 6.2. Weight Loss Surgery

Currently, the only available effective treatment for extreme obesity (BMI ≥ 40) and metabolic syndrome is weight loss surgery (WLS). Individuals with WLS experience body weight loss between 20% and 30% and the alleviation of comorbidities, such as type 2 diabetes, at over 80% [150]. WLS is on the rise worldwide, with the two most common types being RYGB and vertical sleeve gastrectomy (VSG), each amounting to more than 200,000 procedures a year [151]. Essentially, RYGB surgery creates an anastomoses between the stomach and the jejunum, where stomach contents can bypass the initial secretions from the proximal small intestine [152]. VSG, on the other hand, involves removing a majority of the greater curvature of the stomach, leaving the entirety of the small intestine intact [153]. The weight-loss that occurs after surgery does not appear to reflect mechanical restriction of food intake or malabsorption of nutrients, but a change in the bi-directional regulatory signaling between the brain and the gut. Accumulating evidence implicates gut microbiota in WLS outcomes [154]. It is well established that gut microbiota differ, in community structure, gene content and metabolic network organization, between obese and lean individuals [155,156], and that it is significantly and persistently changed by WLS [157]. It has been shown that changes in gut microbiota causally contributes to reduced weight and adiposity after WLS [158].

While current literature remains diverse about the specific changes in the composition of the microbiota following WLS, they are likely due to multiple factors, including, but not limited to, species differences (i.e., human versus murine models), concomitants from diet regimens, and high variability in the time course for sampling or disease state [159,160]. Some potentially important observations have been reported, some of which are independent of the surgery type [157], and some that are specific to the procedure [161]. For example, one study found that VSG and RYGB differed in their effect on microbiota composition profiles, with higher levels of *Akkermansia, Eubacterium*, *Haemophilus*, and *Blautia* for SG, while *Veillonella*, *Slackia*, *Granucatiella,* and *Acidaminococcus* were present in greater levels in RYGB. RYGB-induced microbiota changes were also reflected at the level of functionality, especially in pathways related to environmental adaptation. Other studies confirming some of the findings appear to be specific to RYGB, such as increased *Akkermansia muciniphilia* and *Faecalibacterium prausnitzii* [162]. Overall, the phyla Gammaproteobacteria have been noted to increase while Firmicutes decrease post-RYGB [163]. On the other hand, a biomarker discovery analysis revealed the genus *Blautia* as characteristic in VSG. One study in rats [164] found the RYGB group was postoperatively enriched for Gammaproteobacteria and Bacteroidaceae, whereas the VSG group was postoperatively enriched for Desulfovibrionaceae and Cyanobacteria. Compared to the pre-operative parameters, the RYGB group had a persistent increase in the relative abundance of Gammaproteobacteria and a decrease in the Shannon index, while the SG group only transiently exhibited these changes within the first week after surgery. The relative abundance of Gammaproteobacteria was negatively correlated, whereas the Shannon index was positively correlated with weight after surgery. Despite the authors conclusion that RYGB, but not VSG, alters the gut microbiota, at least in Sprague-Dawley rats, a recent study [165] showed marked rescue of high fat diet effects on the composition of gut microbiome following VSG in Sprague-Dawley rats. Specifically, Firmicutes/Bacteroidetes ratio was restored by VSG to the level of normal diet controls, despite that the VSG rats remained on HFD. These pre-clinical data are promising, as they show the potency of VSG to improve parameters of gut dysbiosis associated with dietary obesity with the potential to promote re-colonization toward a healthier gut microbiota, one that is richer in Bacteroidetes and SCFAs.

Considering that PD is caused by neurodegeneration of dopaminergic neurons and resulting reductions in dopamine bioavailability, it is plausible to assume that a procedure increasing dopamine levels can serve as a potential treatment for the disease. In a recent study, it was shown that one of the long-term effects of weight loss from the RYGB procedure in obese women is an increase in D2/D3 receptor availability in the striatum (Figure 4) [22]. Striatal dopamine receptor expression is correlated with dopamine bioavailability. The authors propose that the increases seen in striatal D2/D3 receptors can be attributed to RYGB-induced changes in gut hormones such as GLP-1. *Faecalibacterium prausnitzii,* shown to be elevated in RYGB patients secretes GLP-1 [166], which was shown to increase DAT expression in the forebrain [130]. In a separate study, high-fat fed rats that underwent RYGB procedure had increased binding to D1/D2 receptors and DAT [137], suggesting that RYGB may restore dopamine deficits related to dietary obesity. These findings are also largely supported by studies demonstrating increased expression of dopamine receptors following WLS. It has been documented that downregulation of the central µ-opioid receptor and D2 receptor, in particular, is implicated in weight gain and obesity [167]. Evidence show that WLS rescue the interaction between the central µ-opioid receptor and D2 receptor in the ventral striatum. Similarly, reduced central µ-opioid receptor post-RYGB surgery was found in brain regions associated with stress and energy regulation such as the amygdala [168]. Furthermore, RYGB-like rerouting of dietary fat products of digestion stimulates satiety through increased vagus nerve driven dorsal striatal dopamine concentrations and upregulated D1R expression [169]. Although the preponderance of the literature supports increased activity and availability of dopamine receptors in the brain post-WLS, conflicting studies exist [170]. However, more recent studies confirmed the positive impact of WLS on dopamine receptor expression.

Changes in gut microbiota composition following RYGB have also been linked with other dopaminergic neural substrates, such as the reward system. For example, GF mice have increased serotonin turnover, noradrenaline, and dopamine in the striatum [171,172]. Further, administration of psychobiotic strains stimulates dopaminergic mesolimbic pathways and improves dopaminergic deficits [173]. Recent animal and human studies show a direct effect of microbiota on modulation of DAT in brain areas known to control reward and food intake. Specifically, microbiota transfer from RYGB patients to metabolic syndrome individuals increased striatal DAT binding and produced changes in microbe profile involved in dopamine synthesis pathways [174]. Further, administration of *Bacteroides uniformis* in a rat model of food addiction improved binge eating, an effect mediated by NAc dopamine and prefrontal cortex D1 and D2 receptors [175]. Finally, *Akkermansia muciniphilia’s* products (e.g., propionate) have been shown to modulate the mesolimbic dopaminergic signaling leading to reduced dopamine release in NAc shell [176]. SCFAs, such as acetate, butyrate and propionate, are produced by gut microbial fermentation of dietary fiber, act via GPCRs (FA2R, FA3R, OLFR78, GPR109A), and play a key role in microbiota–gut–brain crosstalk [177]. In fact, recent studies have demonstrated the efficacy of propionate-producing consortium to restore antibiotic-induced dysbiosis in humans [178] and regulate metabolic, inflammatory, and neural pathways [179,180]. Taken together, although emerging evidence supports positive effects on gut microbiota, dopamine bioavailability and satiety following WLS, direct evidence linking WLS to improvements in diseases such as PD remain to be seen.

## 7. Conclusions and Perspectives

Substantial evidence supports the role of nutrition on restructuring gut microbiota in ways that increase susceptibility to, or confer neuroprotection against, PD. More specifically, dietary proteins, fats and carbohydrates, as part of the MD and WD, have individual, critical and varying roles in dopaminergic pathways, as well as dopamine metabolism and bioavailability. Foods rich in whole-grain cereals, wheats, nuts, seeds, PUFA, fruits and vegetables characteristic of MD, and those rich in fats, sugars and animal proteins that comprise the WD, have inverse effects on gut microbe composition and risks of PD development. Further, microbial sequencing of the parkinsonian gut has shown similarities to diet-specific changes induced by the MD and WD. For example, the Western diet induces decreases in *Prevotella, Roseburia* and *Eubacterium* spp., while increasing *Ruminococcus* spp. [94,108,109], while the Mediterranean diet has opposite effects. Overall, Western Diet-induced changes stimulate a systemic inflammatory state primarily through chronic LPS-mediated endotoxemia leading to excessive microglial inflammation [95]. Thus, microglia promote neuroinflammation via production of IL-1 and TNF-α that results in irreversible CNS damage. Similarly, the WD is shown to be strongly associated with obesity and other metabolic diseases that induces changes in resident gut bacteria, also promoting neuroinflammation through microgliosis and astrogliosis [181].

Our review highlights therapeutic modalities that can be used to restore beneficial gut microbial concentrations, and attenuate dopaminergic and PD deficits. As such, FMT restores striatal dopamine concentrations, reduces dopaminergic neuronal loss, and decreases LPS/microglial induced neuroinflammation. Additionally, FMT is able to restore the effects of p-cresol which is a key proteolytic byproduct of the WD. WLS, on the other hand, has been shown to treat obesity and reduce resulting comorbidities by causing alterations in the gut microbiota to affect gut-brain signaling and the dopaminergic signaling pathways. WLS restores dopamine deficits by increasing dopaminergic receptor and transporter expression. RYGB surgery also increases levels of ghrelin and GLP-1, which have been positively associated with obesity, dietary interventions, and dopamine production [182]. The specific mechanisms on how the complex microbiota and/or the abundance or absence of single bacterial strains impact the dopaminergic system in the gut and the brain have been recently reviewed elsewhere [14].

Although significant progress has been made in elucidating the mechanisms by which therapeutic modalities such as FMT and WLS that change gut microbial composition can influence dopamine-related pathologies, there remains much to be learned. PD is a multifactorial disorder that involves a combination of systemic, genetic, and environmental factors. One limitation of FMT is that nearly all studies that show restoration of dopamine functions or alleviation of PD deficits were conducted using animal models with Parkinsonian features through MPTP or rotenone-induced neurotoxicity. Similarly, some studies used donor feces from humans that were transplanted into recipient animal models [20], which raises questions related to host intestinal interactions with donor microbiota due to species-specific differences in the rodent and human microbiota [183]. Although the results from animal studies are promising, the benefits of FMT, as shown through human studies, to date, are mainly limited to the symptomatic relief of intestinal dysmotility and motor symptoms commonly experienced by PD patients. One recent case series of six patients that underwent FMT via colonoscopy reported improvements in PD symptoms with no adverse events [184]. In another preliminary study, with 15 study participants, it was shown that FMT reduced motor symptoms as measured by the Unified Parkinson’s Disease Rating Scale III (UPDRS) at 1 month and 3 month follow-up [185]. However, it should be noted that some patients in this study reported mild, self-limited adverse effects after 1 week, such as diarrhea, abdominal pain and flatulence. Another, more recent study of 11 PD patients demonstrated improvements in UPDRS scores post-FMT treatment, while reducing Bacteroidetes and increasing *Blautia* and *Prevotella* species [186]. Again, similar mild and self-limiting side effects were reported; however, no serious adverse effects were noted in any of the above studies. Still, the combined sample sizes within these studies remain small. As such, studies with increased statistical power such as randomized control trials need to be conducted in humans to assess long-term effects and safety prior to regular use of this therapeutic modality in PD.

WLS, on the other hand, has its own limitations. Although clear associations have shown the beneficial effects of gut microbiota and resulting effects on key gut hormones, dopamine transporters and receptors, WLS has not been well studied in direct association with PD. Several studies have focused more largely on the positive effects of WLS on reward pathways, specifically the hedonic control of food consumption. It has been shown that WLS-induced increases in striatal dopamine transmission can improve reward sensitivity, hormonal signaling and learned food avoidances [187]. This focus has contributed to the limited direct evidence of WLS on PD available in the current literature. However, the link between obesity, insulin regulation and PD suggests that WLS can serve as a promising therapeutic method to delay onset of PD by reducing systemic and neuroinflammation through alterations in gut microbiota. Since WLS has become increasingly common as a treatment for obesity, prospective or retrospective studies evaluating the long-term effects of WLS on the onset or delay of PD pathogenesis are indicated.

It is also worth pointing out that specific changes in microbiota composition post-WLS can vary based on species differences (i.e., human versus murine models), diet regimens, and time-course, all of which can affect its association with key aspects of the dopaminergic pathway.

Nevertheless, the preliminary data linking WLS, FMT and dopamine is promising, and future studies should focus on further elucidation of the mechanisms by which these therapeutic targets can prevent, delay or restore PD deficits.

## Figures and Tables

**Figure 1 ijms-23-07503-f001:**
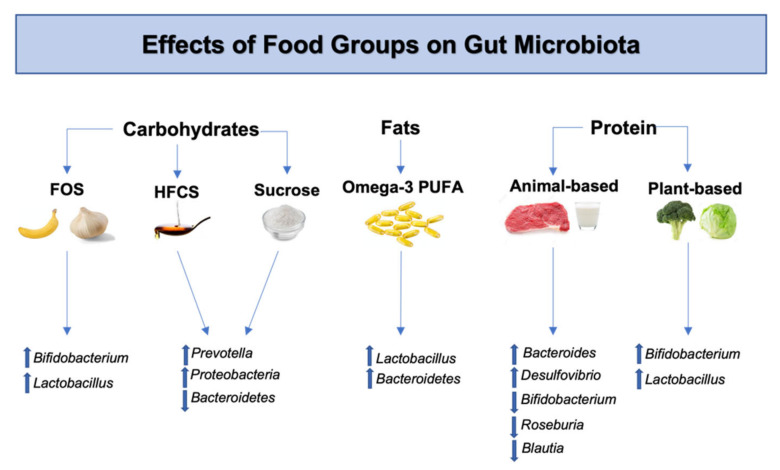
Effects of food groups on gut microbiota. FOS increase *Bifidobacterium* and *Lactobacillus* species in the gut [35]. HFCS and sucrose administration increases *Prevotella* and Proteobacteria while decreasing Bacteroidetes [36]. Omega-3 PUFA increase *Lactobacillus* and *Bacteroidetes* [37]. Animal-based proteins increase *Bacteroides* and *Desulfovibrio* while decreasing *Bifidobacterium, Roseburia* and *Blautia* [38,39]. Plant-based proteins increase *Bifidobacterium* and *Lactobacillus* [40]. Abbreviations: FOS, fructo-oligosaccharides; HFCS, high-fructose corn syrup; PUFA, poly-unsaturated fatty acids.

**Figure 2 ijms-23-07503-f002:**
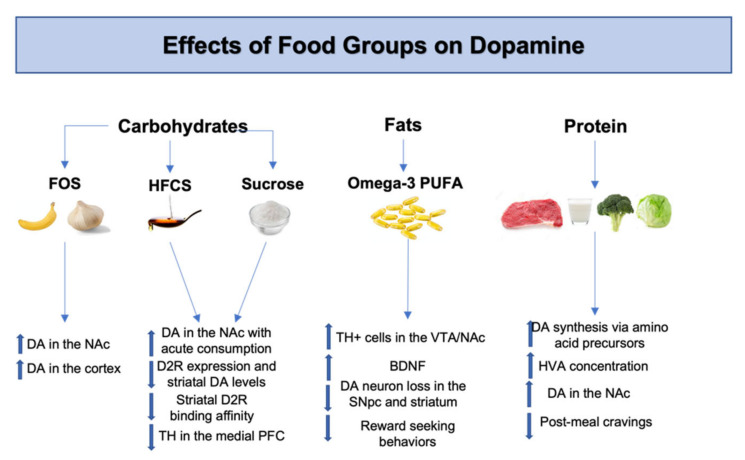
Effects of food groups on dopamine. FOS consumption increases dopamine levels in the NAc and the cortex [46,47]. Acute HFCS and sucrose consumption increases dopamine in the NAc [48]. Chronic HFCS and sucrose consumption decreases D2R expression, D2R binding affinity, striatal dopamine levels and TH in the medial PFC [50,52]. Omega-3 PUFA increase BDNF and TH+ cells in the VTA and NAc [57]. Omega-3 PUFA decrease dopamine neuron loss in the SNpc and striatum [58], while also reducing reward seeking behavior [59]. Protein intake increases dopamine synthesis, dopamine levels in the NAc and HVA concentrations while decreasing post-meal cravings [60,61,62]. Abbreviations: FOS, fructo-oligosaccharides; HFCS, high-fructose corn syrup; PUFA, poly-unsaturated fatty acids, DA, dopamine; NAc, nucleus accumbens; D2R, dopamine 2 receptor; TH, tyrosine hydroxylase; PFC, pre-frontal cortex; BDNF, brain derived neurotrophic factor; VTA, ventral tegmental area; SNpc, substantia nigra pars compacta; HVA, homovanillic acid.

**Figure 3 ijms-23-07503-f003:**
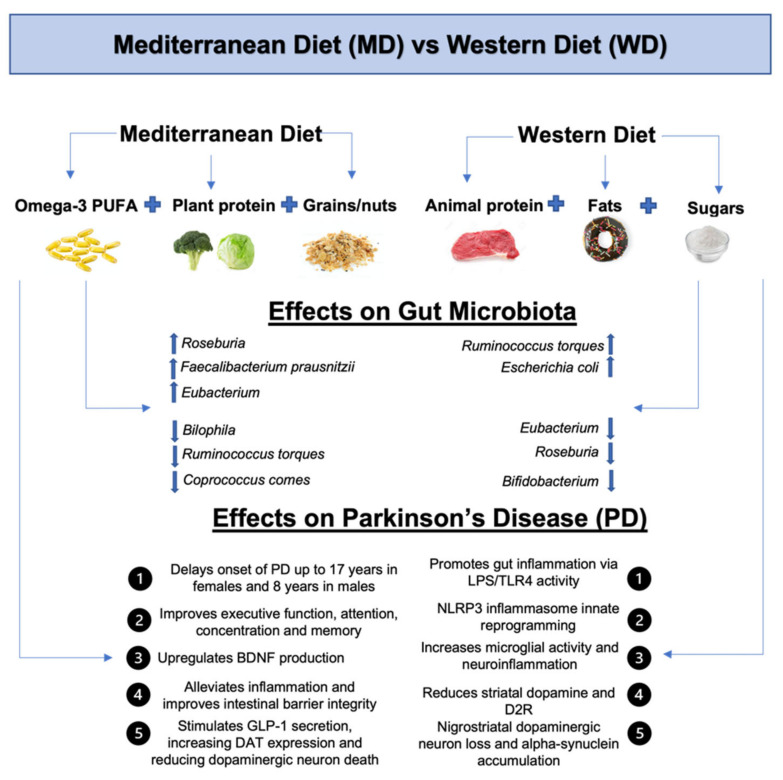
The effects of the Mediterranean Diet and Western Diet on gut microbiota and Parkinson’s Disease. The MD is composed of Omega-3 PUFA, plant protein, grains, and nuts. The MD increases concentrations of *Roseburia, Faecalibacterium prausnitzii* and *Eubacterium,* while lowering concentrations of *Bilophila, Ruminococcus torques, Coprococcus comes.* The MD effects on PD include delaying onset, improving executive function in PD patients, upregulating BDNF production, alleviating inflammation, improving intestinal barrier integrity and stimulating GLP-1 secretion, which in turn increases DAT expression and reduces dopaminergic neuron death. The WD is composed of animal proteins, fats and sugars. The WD increases concentrations of *Ruminococcus torques* and *Escherichia coli,* while reducing levels of *Eubacterium, Roseburia* and *Bifidobacterium* in the gut. The WD effects on PD include promoting gut inflammation via LPS/TLR4 activity, NLRP3 inflammasome innate reprogramming, increasing neuroinflammation, reducing striatal dopamine and D2R, leading to nigrostriatal dopaminergic loss and α-synuclein accumulation. Abbreviations: PUFA, poly-unsaturated fatty acids; BDNF, brain derived neurotrophic factor; GLP-1, glucagon-like peptide 1; DAT, dopamine transporter; LPS, lipopolysaccharides; TLR-4, toll-like receptor 4; D2R, dopamine 2 receptor.

**Figure 4 ijms-23-07503-f004:**
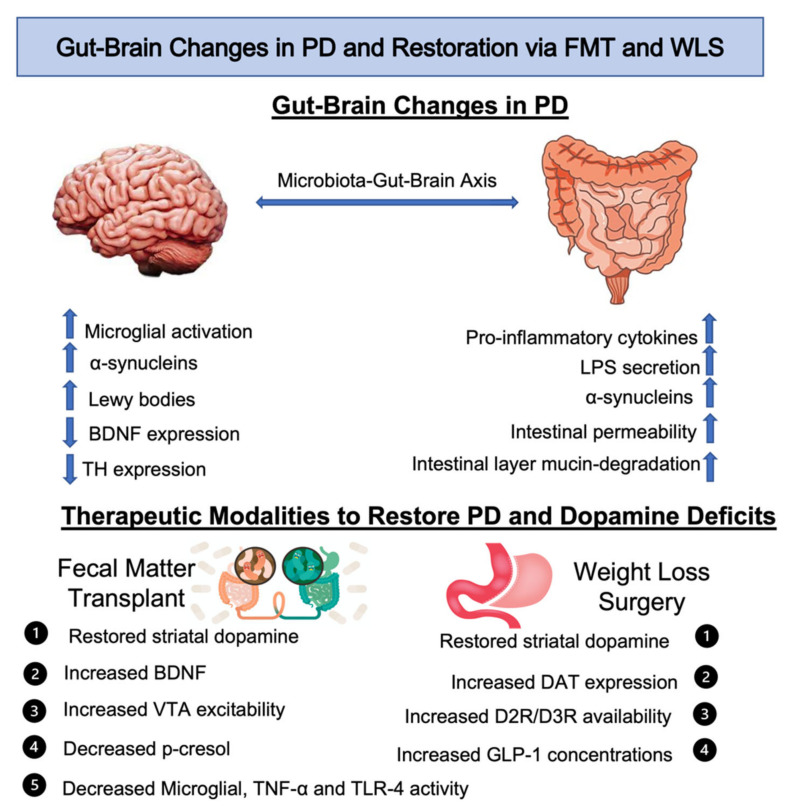
Gut-brain changes in Parkinson’s Disease and restoration of dopaminergic and Parkinson’s disease deficits via fecal microbiota transplant and weight loss surgery. Changes in the brain include decreased BDNF levels, reduced tyrosine hydroxylase expression, increased α-synuclein and microglial activity. Changes in the gut include increased intestinal permeability, mucin-degradation and α-synucleins, LPS secretion and pro-inflammatory cytokines. Fecal microbiota transplant increases striatal dopamine concentrations, VTA dopamine excitability and BDNF, while decreasing p-cresol levels and microglial activity. Weight loss surgery increases DAT expression, D2R/D3R expression, GLP-1 levels, and striatal dopamine concentrations. Abbreviations: BDNF, brain derived neurotrophic factor; TH, tyrosine hydroxylase; LPS, lipopolysaccharide; TNF-α, tumor necrosis factor α; TLR-4, toll-like receptor 4; VTA, ventral tegmental area; DAT, dopamine transporter; D2R, dopamine 2 receptor; D3R, dopamine 3 receptor; DAT, dopamine transporter; GLP-1, glucagon-like peptide 1; FMT, fecal microbiota transplant; WLS, weight loss surgery.

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
