# Peer review of "Impact of Nutrition, Microbiota Transplant and Weight Loss Surgery on Dopaminergic Alterations in Parkinson’s Disease and Obesity"

_ijms, 2022, doi:10.3390/ijms23147503_

Round 1
Reviewer 1 Report
In their review article, the authors summarize recent findings on the effects of nutrients, dietary interventions, fecal microbiota transplant and weight loss surgery on the intestinal microbiota and associated changes in the dopaminergic system. They graphically present the current scientific knowledge in several illustrations that show, to some extent, connections between the gut microbiota and dopaminergic changes in Parkinson’s disease (PD) and obesity, and effects of fecal microbiota transplant (FMT) and weight loss surgery (WLS). Overall, the manuscript is written clearly and comprehensively, and is based on current research findings in rodents and humans. However, there is some similarity with a review of the authors recently published in MDPI Biomedicines. While it is on the one hand important to explain main differences between the two publications and the novelty of the submitted review, some relevant references discussed in the previous review might also be added to the current article. Overall, it is recommended to revise and clarify some important aspects prior to publication.
Major points:
1. Table 1: The informative value of the table is low, as the listed findings already included in the main text, and summarized in Figure 4. It is assumed that the absence as well as the presence of specific bacterial strains have effects on the dopaminergic system. It is thus recommended to provide more data and information in the table. The title of the table states effects of microbes on dopaminergic pathways via gut microbiota manipulations. However, the table comprises overall changes in the gut and brain, and does not enlarge on specific pathways. Further, there is no information on how the FMT and WLS manipulate the gut microbiota. It is recommended to change the title of the table to be more specific, such as: “Microbe-associated effects of microbiota transplant and weight loss surgery on dopaminergic pathways”.
2. Figure 4 has strong similarity with Figure 3 of a recently published review on the role of microbiota-gut-brain axis in regulating dopaminergic signaling (Hamamah et al., Biomedicines, 2022). Please cite the reference in the figure legend and redesign Figure 4 to make it more different from the Figure used in the other review. In general, please provide a clear-cut distinction of the two reviews and avoid self-plagiarism.
Moreover, the style if Figure 4 should become more homogenous and cartoon-like elements or caricatures should be avoided. Small spaces between epithelial cells could visually highlight “intestinal permeability”. Instead of single primary protein structures, genuine α-synuclein-containing lewy bodies could be shown. It is recommended to depict gut-brain changes in PD on one side of the gut/brain and outcomes of therapeutic interventions on the other side. Use frames and colored backgrounds to clarify allocations. The line connecting the brain and the gut needs to be labeled accordingly.
Minor points:
1. The heading emphasizes the positive effects of FMT and WLS on dopaminergic deficits in PD and obesity. In the review, however, the main focus lies on nutrition- and nutrient-related effects on the microbiota and dopaminergic system, whereas FMT and WLS are solely addressed in the last quarter of the text. In addition, the association between FMT or WLS and the dopaminergic system is mainly based on experimental findings in mice and causal evidence in humans is lacking. The heading of the review is thus considered misleading and it is suggested to be changed, for example: “Impact of nutrition, microbiota transplant and weight loss surgery on dopaminergic alterations in Parkinson’s disease and obesity” or “Modulation of dopaminergic alterations in Parkinson’s disease and obesity by dietary interventions, fecal microbiota transplant and weight loss surgery”.
2. In section 6.1., the text highlights the potential beneficial effects of FMT on dopaminergic alterations, but does not mention adverse effects of FMT in PD patients (Xue et al., Medicine, 2020). Please discuss possibilities, but also limitations and risks of FMT.
3. In section 6.2., effects of WLS are reviewed. Readers addressed by the journals review include non-clinical scientist that are not familiar with different WLS methods. Please introduce the described WLS methods, so the reader is familiar with the procedures.
4. Figure 3/Figure 4: There are two Figures named “Figure 3”. Please name the later one “Figure 4”.
5. In section 7., please discuss the mechanisms how the complex microbiota and/or the abundance or absence of single bacterial strains impacts the dopaminergic system in the gut and the brain more thoroughly. Provide an outlook and mention current clinical trials on FMT and WLS in neurological, dopamine-related disorders such as PD.
6. Please be consistent with abbreviations and italics throughout the text. Consult the guidelines on scientific nomenclature (cdc.gov) for bacteria names and italics.
7. Lines 177, 181, 435: The abbreviation for TNF-alpha is provided in line 181, but the abbreviated form is already used in line 177. Starting from line 435, TNF-alpha is abbreviated TNF-α. Please use uniform spelling.
8. Lines 229 and 328: The abbreviation for LPS is indicated in line 328, but the abbreviated form is already used in line 229.
9. Line 233: The full name of the enzyme MAO-B and a brief introduction about its functions may be useful, as the dopaminergic pathway is not addressed throughout the review.
10. Line 244: Replace “Fo” by “For”.
11. Lines 246, 249, 373: Replace “species” by “species”.
12. Lines 249-250: Please add a reference or revise the sentence, as the development of inflammatory bowel states is complex and not alone based on the abundance of a certain bacterial consortium.
13. Line 303: Replace “The” by “The”.
14. Line 309: Replace “significant uptakes in” by “significantly upregulated”.
15. Line 318: Replace “Coli” by “coli”, and “diets” by “diet”.
16. Line 347: Add the full name for “HPA” and some information on its function.
17. Lines 356, 361, 407, 410: Replace “and” by “and”.
18. Line 361: Replace “in the gut” by “in the gut”.
19. Line 371: Add a comma between “Akkermansia Eggerthella”.
20. Line 402: Starting from line 402, alpha-synuclein is abbreviated α-synuclein. Please use uniform spelling.
21. Lines 402, 443, 458: When using the plural form “α-synucleins”, do you mean different conformation or aggregation states, or (non)pathological forms? Please specify or use the singular form.
22. Line 460: It is not quite clear which pathway is meant in particular. Please be specific.
23. Line 487: Replace “benedicial” by “beneficial”.
24. Line 505: Please add the abbreviation for Roux-en-Y gastric bypass used hereinafter.
25. Line 518: Replace “Clostridoides” by “Clostridioides”.
26. Line 525, 526: Please use uniform spelling for p-cresol/P-cresol.
27. Line 531: The text says: “FMT of normal microbiota into the p-cresol treated mice normalized p-cresol” (mediated effects?). Please add missing words.
28. Line 532: The reader may not be familiar with “EphB6-deficient mice”. Please provide information on the mouse model to better understand the context and meaning of the statement.
29. Line 560: Replace “IL-B1” by “IL-1β”.
Author Response
In their review article, the authors summarize recent findings on the effects of nutrients, dietary interventions, fecal microbiota transplant and weight loss surgery on the intestinal microbiota and associated changes in the dopaminergic system. They graphically present the current scientific knowledge in several illustrations that show, to some extent, connections between the gut microbiota and dopaminergic changes in Parkinson’s disease (PD) and obesity, and effects of fecal microbiota transplant (FMT) and weight loss surgery (WLS). Overall, the manuscript is written clearly and comprehensively, and is based on current research findings in rodents and humans. However, there is some similarity with a review of the authors recently published in MDPI Biomedicines. While it is on the one hand important to explain main differences between the two publications and the novelty of the submitted review, some relevant references discussed in the previous review might also be added to the current article. Overall, it is recommended to revise and clarify some important aspects prior to publication.
Thank you for your constructive and positive comments which helped improve our paper
Major points:
- Table 1: The informative value of the table is low, as the listed findings already included in the main text, and summarized in Figure 4. It is assumed that the absence as well as the presence of specific bacterial strains have effects on the dopaminergic system. It is thus recommended to provide more data and information in the table. The title of the table states effects of microbes on dopaminergic pathways via gut microbiota manipulations. However, the table comprises overall changes in the gut and brain, and does not enlarge on specific pathways. Further, there is no information on how the FMT and WLS manipulate the gut microbiota. It is recommended to change the title of the table to be more specific, such as: “Microbe-associated effects of microbiota transplant and weight loss surgery on dopaminergic pathways”.
- Figure 4 has strong similarity with Figure 3 of a recently published review on the role of microbiota-gut-brain axis in regulating dopaminergic signaling (Hamamah et al., Biomedicines, 2022). Please cite the reference in the figure legend and redesign Figure 4 to make it more different from the Figure used in the other review. In general, please provide a clear-cut distinction of the two reviews and avoid self-plagiarism.
Moreover, the style if Figure 4 should become more homogenous and cartoon-like elements or caricatures should be avoided. Small spaces between epithelial cells could visually highlight “intestinal permeability”. Instead of single primary protein structures, genuine α-synuclein-containing lewy bodies could be shown. It is recommended to depict gut-brain changes in PD on one side of the gut/brain and outcomes of therapeutic interventions on the other side. Use frames and colored backgrounds to clarify allocations. The line connecting the brain and the gut needs to be labeled accordingly.
Response: We have reconstructed figure 4 to be much different from how it appears in the previously published paper and created it in a similar format that the other figures in this current paper appear. As such, we believe that the two figures are now very dissimilar. Further, we agree that table 1 does repeat much of the information that is already in figure 4 and therefore removed the table to avoid repetition.
Minor points:
- The heading emphasizes the positive effects of FMT and WLS on dopaminergic deficits in PD and obesity. In the review, however, the main focus lies on nutrition- and nutrient-related effects on the microbiota and dopaminergic system, whereas FMT and WLS are solely addressed in the last quarter of the text. In addition, the association between FMT or WLS and the dopaminergic system is mainly based on experimental findings in mice and causal evidence in humans is lacking. The heading of the review is thus considered misleading and it is suggested to be changed, for example: “Impact of nutrition, microbiota transplant and weight loss surgery on dopaminergic alterations in Parkinson’s disease and obesity”or “Modulation of dopaminergic alterations in Parkinson’s disease and obesity by dietary interventions, fecal microbiota transplant and weight loss surgery”.
Response: We appreciate the title recommendation and have changed it to “Impact of nutrition, microbiota transplant and weight loss surgery on dopaminergic alterations in Parkinson’s disease and obesity”
- In section 6.1., the text highlights the potential beneficial effects of FMT on dopaminergic alterations but does not mention adverse effects of FMT in PD patients (Xue et al., Medicine, 2020). Please discuss possibilities, but also limitations and risks of FMT.
Response: Thank you for providing this reference. We have added this study to the paper and mentioned the adverse effects that were reported by the authors of that study.
- In section 6.2., effects of WLS are reviewed. Readers addressed by the journals review include non-clinical scientist that are not familiar with different WLS methods. Please introduce the described WLS methods, so the reader is familiar with the procedures.
Response: We explained the general methodology of both RYGB and VSG in that section.
- Figure 3/Figure 4: There are two Figures named “Figure 3”. Please name the later one “Figure 4”.
Response: The figures were renamed accordingly
- In section 7., please discuss the mechanisms how the complex microbiota and/or the abundance or absence of single bacterial strains impacts the dopaminergic system in the gut and the brain more thoroughly. Provide an outlook and mention current clinical trials on FMT and WLS in neurological, dopamine-related disorders such as PD.
Response: The discussion on specific mechanisms by which microbiota impact the dopaminergic system has been the subject of our recently published review in Biomedicines 2022, 10(2), 436; https://doi.org/10.3390/biomedicines10020436. To avoid duplication, we’ve cited the paper in the discussion section.
- Please be consistent with abbreviations and italics throughout the text. Consult the guidelines on scientific nomenclature (cdc.gov) for bacteria names and italics.
Response: Thank you for providing this feedback. We went through the manuscript to italicize the bacteria names in accordance with the scientific guidelines
- Lines 177, 181, 435: The abbreviation for TNF-alpha is provided in line 181, but the abbreviated form is already used in line 177. Starting from line 435, TNF-alpha is abbreviated TNF-α. Please use uniform spelling.
Response: We corrected this accordingly throughout the text
- Lines 229 and 328: The abbreviation for LPS is indicated in line 328, but the abbreviated form is already used in line 229.
Response: We corrected this accordingly throughout the text
- Line 233: The full name of the enzyme MAO-B and a brief introduction about its functions may be useful, as the dopaminergic pathway is not addressed throughout the review.
Response: We added a line about the role of MAO-B in dopamine metabolism and how it pertains to the study where it was introduced.
- Line 244: Replace “Fo” by “For”.
Response: We made this correction.
- Lines 246, 249, 373: Replace “species” by “species”.
Response: These changes were addressed.
- Lines 249-250: Please add a reference or revise the sentence, as the development of inflammatory bowel states is complex and not alone based on the abundance of a certain bacterial consortium.
Response: We agree with this comment. We revised the sentence by adding that it contributes to inflammatory bowel states versus causing it alone.
- Line 303: Replace “The” by “The”.
Response: This change was addressed.
- Line 309: Replace “significant uptakes in” by “significantly upregulated”.
Response: This change was addressed.
- Line 318: Replace “Coli” by “coli”, and “diets” by “diet”.
Response: This change was addressed.
- Line 347: Add the full name for “HPA” and some information on its function.
Response: We added the full name for HPA as well as how HPA axis dysregulation is connected with gut microbial alterations.
- Lines 356, 361, 407, 410: Replace “and” by “and”.
Response: These changes were addressed.
- Line 361: Replace “in the gut” by “in the gut”.
Response: This change was addressed.
- Line 371: Add a comma between “Akkermansia Eggerthella”.
Response: This change was addressed.
- Line 402: Starting from line 402, alpha-synuclein is abbreviated α-synuclein. Please use uniform spelling.
Response: This change was addressed.
- Lines 402, 443, 458: When using the plural form “α-synucleins”, do you mean different conformation or aggregation states, or (non)pathological forms? Please specify or use the singular form.
Response: These changed were addressed. Two of them were changed to the singular form while one was changed to alpha-synuclein aggregation states
- Line 460: It is not quite clear which pathway is meant in particular. Please be specific.
Response: This sentence was edited to become more clear
- Line 487: Replace “benedicial” by “beneficial”.
Response: This change was addressed.
- Line 505: Please add the abbreviation for Roux-en-Y gastric bypass used hereinafter.
Response: This change was addressed.
- Line 518: Replace “Clostridoides” by “Clostridioides”.
Response: This change was addressed.
- Line 525, 526: Please use uniform spelling for p-cresol/P-cresol.
Response: This change was addressed. We uniformly changed it to p-cresol
- Line 531: The text says: “FMT of normal microbiota into the p-cresol treated mice normalized p-cresol” (mediated effects?). Please add missing words.
Response: We added the missing words (mediated effects)
- Line 532: The reader may not be familiar with “EphB6-deficient mice”. Please provide information on the mouse model to better understand the context and meaning of the statement.
Response: This change was addressed. We added the EphB6 association as a candidate gene for Autism Spectrum Disorder
- Line 560: Replace “IL-B1” by “IL-1β”.
Response: This change was addressed.
Reviewer 2 Report
Overall, the study is well designed and well performed. All the sections are clearly presented and discussed. The subject is highly critical and accurate,
Although the flaws within the manuscript, I suggest its publication in case of minor revision. Some indications for minor revision are given below.
Try to develop the introduction with few lines about probiotics and beneficial microorganisms and their involvement in mental disorder regulation.
In addition to FMT and Weight Loss Surgery , are there any other strategies to restore Dopamine and PD deficits ?
Are there any other studies or investigations that underscore the role of probiotics as modulators and/or regulators of Parkinson’s Disease, its symptoms, notably these beneficial microbes have shown a great potential in the field of modulation of neurodegenerative disorders?
Line 81: "It is also known that gut microbiota play significant roles..". It shoud be modified as "It is also known that gut microbiota plays significant roles...".
Line 243: "However, the effects of derived metabolites is dependent...". You should modify it as follows: "However, the effects of derived metabolites are dependent...".
Line 290-291: Try to use one style of writing.
Line 303: "The implications...". Try to remove the italics in "The".
Line 305: "Gut microbota...". Modify it as "Gut microbiota...".
Is there any future outlook about the use of WLS and/or FMT in treating of PD and dopamine deficits, not only at research scale, but at clinical or medical scale?
All the figures are well presented and explained.
Check the english style all through the manuscript.
Check all the text in order to put commas and punctuation in right places to give the meaning to the sentences and ideas.
Verify all the text for italic mode.
Author Response
Overall, the study is well designed and well performed. All the sections are clearly presented and discussed. The subject is highly critical and accurate,
Although the flaws within the manuscript, I suggest its publication in case of minor revision. Some indications for minor revision are given below.
Thank you for your constructive and positive comments which helped improve our paper
Try to develop the introduction with few lines about probiotics and beneficial microorganisms and their involvement in mental disorder regulation? Are there any other studies or investigations that underscore the role of probiotics as modulators and/or regulators of Parkinson’s Disease, its symptoms, notably these beneficial microbes have shown a great potential in the field of modulation of neurodegenerative disorders? In addition to FMT and Weight Loss Surgery , are there any other strategies to restore Dopamine and PD deficits ?
Response: We developed the introduction with a few lines about probiotics, mentioning that it is the most well-described non-pharmacological therapeutic method in the literature as it pertains to restoring dopaminergic deficits in neurodegenerative disease. However, we wanted to focus on FMT/WLS throughout the rest of the review as it is less described and many reviews have already discussed the role of probiotics, while there isn’t nearly as much literature on FMT or WLS.
Line 81: "It is also known that gut microbiota play significant roles..". It should be modified as "It is also known that gut microbiota plays significant roles...".
Response: Thank you for this recommendation This change was made accordingly.
Line 243: "However, the effects of derived metabolites is dependent...". You should modify it as follows: "However, the effects of derived metabolites are dependent...".
Response: Thank you for this recommendation. This change was made accordingly.
Line 303: "The implications...". Try to remove the italics in "The".
Response: Italics were removed.
Line 305: "Gut microbota...". Modify it as "Gut microbiota...".
Response: Thank you for noticing this. The change was made.
Is there any future outlook about the use of WLS and/or FMT in treating of PD and dopamine deficits, not only at research scale, but at clinical or medical scale?
Response. This information has been presented in the discussion section, line 791 through 816.
Round 2
Reviewer 1 Report
The authors addressed my concerns sufficiently. Importantly, the last figure was improved and a better distinction from their previous publication is now possible. I have no further concerns regarding publication in IJMS.